



# Deep learning for monthly rainfall-runoff modelling: a comparison
# with classical rainfall-runoff modelling across Australia
Stephanie Clark[1] , Julien Lerat[1], Jean-Michel Perraud[1], Peter Fitch[1]
[1]CSIRO, Environment, Canberra, ACT, Australia
*Correspondence to*: Stephanie Clark (stephanie.clark@csiro.au)
## Abstract
A deep learning model designed for time series predictions, the long short-term memory (LSTM)
architecture is regularly producing reliable results in local and regional rainfall-runoff applications
around the world. Recent large-sample-hydrology studies in North America and Europe have shown the
LSTM to successfully match conceptual model performance at a daily timestep over hundreds of
catchments. Here we investigate how these models perform in producing monthly runoff predictions in
the relatively dry and variable conditions of the Australian continent. The monthly timestep matches
historic data availability and is also important for future water resources planning, however it provides
significantly smaller training data sets than daily time series. In this study, a continental-scale
comparison of monthly deep learning (LSTM) predictions to conceptual rainfall-runoff model
(WAPABA) predictions is performed on almost 500 catchments across Australia with performance
results aggregated over a variety of catchment sizes, flow conditions, and hydrological record lengths.
The study period covers a wet phase followed by a prolonged drought, introducing challenges for making
predictions outside of known conditions - challenges that will intensify as climate change progresses.
The results show that LSTMs matched or exceeded WAPABA prediction performance for more than
two-thirds of the study catchments; the largest performance gains of LSTM versus WAPABA occurred
in large catchments; the LSTM models struggled less to generalise than the WAPABA models (eg.
making predictions under new conditions); and catchments with few training observations due to the
monthly timestep did not demonstrate a clear benefit with either WAPABA or LSTM.
**Key words [6 max]:** Hydrology and water resources, machine learning, deep learning, benchmarking,
neural networks, process-based modelling
**Major points**
1. A deep learning model (single-layer LSTM) matched or exceeded performance of a WAPABA
rainfall-runoff model in 69% of study catchments.
2. Monthly datasets contain enough information to train the LSTMs to this level.
3. WAPABA struggled in more catchments to make predictions under dry conditions after being
trained on wet conditions than the LSTM did.



## 1. Introduction

With progressively variable climate conditions and the ever-increasing accessibility of hydrologic data,
there comes the opportunity to reconsider how available data is being used to efficiently predict
streamflow runoff on a large scale. Hydrological researchers are increasingly turning to emerging
machine learning techniques such as deep learning to analyse this increasing volume of data, due to the
relative ease of extracting useful information from large datasets and producing accurate predictions
about future conditions without the need for detailed knowledge about the underlying physical systems.
In some cases, machine learning models have been found capable of obtaining more information from
hydrological datasets than is abstracted with traditional models, due to their automatic feature
engineering and ability to effectively capture high-dimensional and long-term relationships (Nearing et
al., 2021, Frame et al., 2021). The continually evolving machine learning field will continue to offer
novel opportunities that can be harnessed for hydrological data analyses, and it is important to understand
how these methods relate to classical models. Here we benchmark a basic machine learning model
against a traditional conceptual model over a large sample of catchments as a step towards a general
understanding of the use of deep learning models as a tool for the task of monthly rainfall-runoff
modelling in Australian catchments.
Deep learning models have been shown in many applications to provide accurate hydrological
predictions and classifications (Shen et al., 2021, Reichstein et al., 2019, Frame et al., 2022). These
models are particularly useful to hydrological studies as they provide the potential to quickly add and
remove predictors (Shen, 2018), scale to multiple catchments (Kratzert et al., 2018, Lees et al., 2021),
automatically extract useful and abstract information from large datasets (Reichstein et al., 2019, Shen,
2018), make predictions in areas with little or no data (Kratzert et al., 2019, Majeske et al., 2022, Ouma
et al., 2022, Choi et al., 2022), and extrapolate proficiently to larger hydrologic events than are seen in
the training dataset (Li et al., 2021, Song et al., 2022).
The long short-term memory network (LSTM, (Hochreiter and Schmidhuber, 1997)), is a deep learning
model that is gaining popularity in hydrology for daily time series predictions at individual basins or
groups of basins due to its ability to efficiently and accurately produce predictions without requiring
assumptions about the physical processes generating the data. The LSTM is a type of recurrent neural
network (RNN). An extension of the multilayer perceptron, the RNN is specifically designed for use
with time series data through its sequential consideration of input data. The LSTM further extends the
RNN to incorporate gates and memory cells, allowing for input data to be remembered over much longer
time periods and for unimportant data to be forgotten from the network. LSTMs make predictions by
taking into account both the short and long temporal patterns in a time series as well as incorporating
information from exogenous predictors. The data-driven detection of intercomponent, spatial and





temporal relationships by these deep learning models can be of particular benefit when attempting to represent systems in which the physical characteristics are not well defined and the intervariable relationships are complex.

The increasing popularity of the LSTM in hydrology is due to its ability to capture the short-term interactions between rainfall and runoff, as well as the long-term patterns and interactions arising from longer-frequency drivers such as climate, catchment characteristics, land use and changing anthropogenic activity. A growing number of publications are applying LSTMs to hydrological simulations and comparing results to process-based or conceptual modelling results.

A gap exists in the literature concerning a comparison of LSTM models and conceptual models at a monthly time step over a large sample of catchments. The conditions in which LSTMs or conceptual models may have an advantage for monthly rainfall-runoff modelling, in a general sense, are not yet understood as most machine learning applications in hydrology are individual-basin case studies (Papacharalampous et al., 2019) at a daily timestep or higher frequency (eg. (Li et al., 2021, Yokoo et al., 2022). Though the LSTM has successfully matched conceptual model performance in a couple large-sample-hydrology studies at daily timesteps (in the USA (Kratzert et al., 2019) and the UK (Lees et al., 2021)) it is yet unknown how these models compare to conceptual models for monthly runoff predictions in relatively dry conditions such as those characterised by Australian catchments.

Monthly hydrological models are important tools for water resources assessments as hydrologic data has historically been recorded at a monthly or longer frequency, and the monthly timestep is often the most practical for water resources planning with many decisions requiring only monthly streamflow predictions. With their simpler structure, fewer parameters and lower data requirements compared to daily models (Hughes, 1995, Mouelhi et al., 2006), monthly models are also useful tools to investigate uncertainty in rainfall-runoff model structure (Huard and Mailhot, 2008) and allow the support of probabilistic seasonal streamflow forecasting systems (Bennett et al., 2017). Due to data availability, models designed to run on monthly timesteps can be used across much larger areas, informing important large-scale water resources decision-making. For these reasons, generalisable models at monthly timesteps are vital. However, the monthly timestep is traditionally a difficult one to model as it requires extracting both short and long-term hydrologic processes (Machado et al., 2011). In a machine-learning context, the monthly time step differs significantly from the daily time step as it drastically reduces the size of the data set available for model training (by a factor of 30). As the convergence of machine learning algorithms typically improves with larger data sets, a central research question of this paper is to explore the capacity of the LSTM algorithm to cope with the reduced amount of input data imposed by the monthly time step.



Some studies have already used the LSTM to model the rainfall-runoff relationship at a monthly time
step in localised studies, showing potential for this application on a broader scale. Ouma et al. (2022)
used monthly aggregated data due to low data availability in three scarcely-gauged basins the Nzoia
River basin, Kenya. Majeske et al. (2022) trained LSTMs with spatially- and temporally-limited data for
three sub-basins of the Ohio River Basin, claiming the daily timestep was superfluous and cumbersome
in some conditions. Lee et al. (2020) found the LSTM adept at preserving long-term memory in monthly
streamflow at a single station on the Colorado River over a 97-year study without any weakening of the
short-term memory structure. Yuan et al. (2018) used a novel method for parameter calibration in an
LSTM for monthly rainfall-runoff estimation at a single station on the Astor River basin in northern
Pakistan. Song et al. (2022) found the LSTM better reproduced observed monthly runoff and simulated
extreme runoff events than a physically-based model at five discharge stations in the Yeongsan River
basin in South Korea.
Large-sample hydrologic studies that assess methods on a large number of catchments are being
increasingly called for in the field of hydrology (Papacharalampous et al., 2019, Mathevet et al., 2020,
Gupta et al., 2014). Papacharalampous et al. (2019) compared the performance of a number of statistical
and machine learning methods (no LSTM) on 2000 generated timeseries and over 400 real-world river
discharge timeseries and determined that the machine learning and stochastic methods provided similar
forecasting results. Mathevet et al. (2020) compared daily conceptual model performance (no machine
learning) for runoff prediction in over 2000 watersheds, determining that performance depended more
on catchment and climate characteristics than on model structure. Kratzert et al. (2018) found individual
daily-scale LSTMs were able to predict runoff with accuracies comparable to a baseline hydrological
model for over 200 differently complex catchments. (Kratzert et al., 2019) found a global LSTM trained
on over 500 basins in the United States with daily data produced better individual catchment runoff
predictions than conceptual and physically-based models calibrated on each catchment individually.
(Lees et al., 2021) produced a global LSTM to model almost 700 catchments in Great Britain, finding
that this model outperformed a suite of benchmark conceptual models, showing particular robustness in
arid catchments and catchments where the water balance does not close. (Jin et al., 2022) compared
machine learning daily rainfall-runoff models to process-based models for over 50 catchments in the
Yellow River Basin in China. (Frame et al., 2021) found that a global LSTM with climate forcing data
performed similarly or outperformed a process-based model on over 500 US catchments, and that in
catchments where hydrologic conditions are not well understood the LSTM was a better choice.
This study aims to determine the ability of a simple machine learning model (a single-layer LSTM) to
match or exceed the performance of a conceptual monthly rainfall-runoff model (the WAPABA model
(Wang et al., 2011)) for predicting runoff using inputs derived from easily accessible climate variables.



A comparison is made on almost 500 basins across Australia, representing a wide variety of catchment
types, hydro-climate conditions, and with differing amounts of historical data. The prediction
performance of the LSTM machine learning models is compared to the WAPABA conceptual models
for each individual catchment. The proportion of catchments in which the runoff prediction performance
of the conceptual model is met or exceeded by the machine learning model is determined. Conditions
under which the machine learning models or the conceptual models may have an advantage are
investigated, such as catchment size, flow level, and length of historical record. The central questions of
this study are:
1) In general, do LSTMs match conceptual model prediction performance on Australian
145       catchments?
2) Is the reduced number of data points due to the monthly time step an issue for training an LSTM?
3) Under what conditions is the LSTM of particular benefit or drawback? (eg. catchment size, flow
148       level, amount of training data, etc.)

The results of this large-sample analysis of LSTM performance over the Australian continent will assist
in understanding whether LSTMs are a justifiable alternative to conceptual models for monthly rainfall-
runoff prediction in Australia and similar environments, including if monthly data sets are sufficient to
produce accurate predictions with the LSTM. Building on these results, further benefits of deep learning
could be harnessed through the creation of larger-scale models that encompass climatic, hydrologic and
anthropogenic patterns spanning multiple catchments, allowing for the sharing of information under
similar conditions and the potential transfer of knowledge between data-rich and data-scarce regions, or
models that blend conceptual models into the machine learning network structure.

## 2. Data and Methods

### 2.1. Data

The catchment and climate data used in this study are from a dataset curated by Lerat et al. (2020)
comprising a selection of basins across Australia. The dataset spans all main climate regions of the
continent, providing data from a variety of rainfall, aridity and runoff regimes, as described in Table 1.
Catchments where some data were marked as suspicious (e.g. high flow data with large uncertainties,
inconsistencies, suspected errors) or with more than 30% missing data were excluded. This left 496
catchments in the study, with locations as shown in Figure 1. The area of the individual catchments ranges
from approximately 5 km$^2$ to 120,000 km$^2$.



Table 1: Characteristics of the study catchments, over the period 1950-2020

| Variable | Min | Q25 | Median | Q75 | Max |
|---|---|---|---|---|---|
| Catchment area (km2) | 4 | 180 | 449 | 1,456 | 119,000 |
| Mean rainfall (mm/y) | 237 | 691 | 887 | 1130 | 3097 |
| Mean PET (mm/y) | 918 | 1280 | 1500 | 1755 | 2321 |
| Mean runoff (mm/y) | 0.5 | 46 | 130 | 275 | 2213 |
| Aridity index rainfall/PET (-) | 0.11 | 0.44 | 0.61 | 0.81 | 2.61 |
| Daily rainfall skewness (-) | 2.4 | 4.8 | 5.9 | 7.4 | 16.7 |
| Runoff coeff. runoff/rainfall (-) | 0.001 | 0.069 | 0.150 | 0.255 | 0.902 |
| % zero flows in daily series | 0.0 | 0.0 | 3.4 | 23.7 | 74.0 |

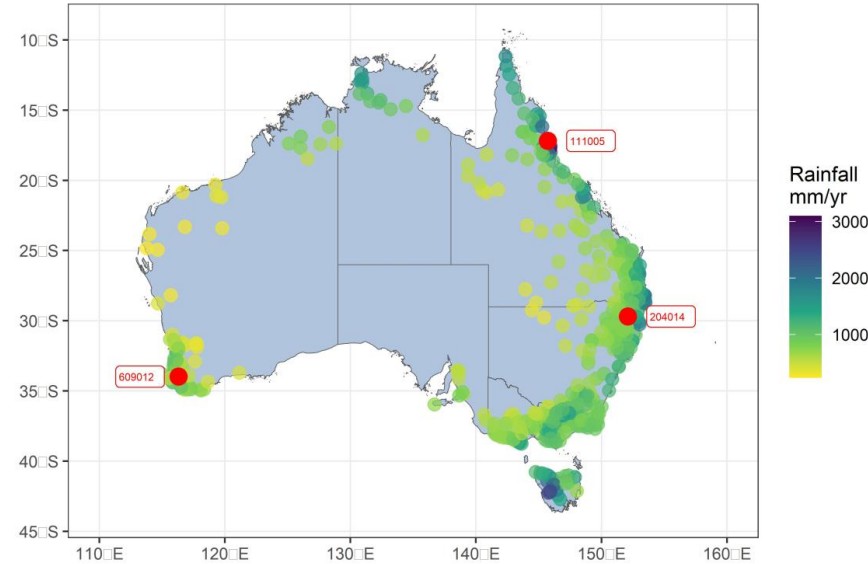


Figure 1: Locations of the 496 study catchments, coloured by mean annual rainfall. The three labelled catchments, which
will be used as examples during the study, represent a wet catchment (111005 in Northern Queensland), a temperate
catchment (204014 in New South Wales), and a dry catchment (609012 in Western Australia).

Observed runoff data were collected from the Bureau of Meteorology's Water Data online portal
(http://www.bom.gov.au/waterdata), rainfall and temperature data are from the Bureau of Meteorology's
AWAP archive (Jones et al., 2009), and potential evapotranspiration data was computed by the Penman
equation as part of the AWRA-L landscape model developed jointly by CSIRO and the Bureau of
Meteorology (Frost et al., 2018). Rainfall, temperature and evapotranspiration are averaged from daily
grids (5x5km) over each of the catchments.





The runoff records begin between January 1950 and September 1982, and end between October 2016
and June 2020. The number of runoff observations per catchment ranges from 425 to 846 with a median
dataset size of 613 observations. The rainfall and potential evapotranspiration data cover the period from
1911 to 2020 continuously. The dataset therefore consists of a set of 496 time series ranging from 37 to
70 years in length, with a median record length of 51 years.

### 186    2.1.1.  Training and testing data split

The data set for each catchment is split into two portions for modelling - in machine learning these are
referred to as 'training' and 'testing' sets, corresponding to the traditional 'calibration' and 'validation'
sets used in hydrologic modelling. The training data set runs from January 1950 (or the start of the
station's record, if later) to December 1995 for all catchments. The testing data set begins in January
1996 for all catchments and ends in July 2020 (or at the end of the station's record, if sooner). This split
is chosen to divide the streamflow records into two relatively even periods, but also to distinguish an
early wet period from a testing period characterised by the Millennium Drought over south-eastern and
eastern Australia (Van Dijk et al., 2013).
When split into training and testing sets at the beginning of January 1996, between 38% and 72% of the
data from each catchment becomes the training set. The length of the training data record for individual
catchments ranges from 14 to 47 years, with the smallest data set used for training containing 172
observations. Typically in machine learning, a portion of the training data is held back to be used during
the model fitting process for monitoring over-fitting and to signal early stopping of training if necessary.
Since the training data sets in this study are already small by machine learning standards, this has not
been done as it would reduce the number of training observations significantly. A sensitivity test has
been performed to justify this choice, and it was found that training the LSTMs with 20% of the training
data reserved for this task produced no apparent benefit in prediction performance.

### 204    2.2.    Models

### 205    2.2.1. Deep learning time series models (LSTMs)

The long short-term memory network, LSTM (Hochreiter and Schmidhuber, 1997), is an updated
recurrent neural network (RNN) specifically designed for deep learning with time series data. The
inclusion of gates and memory cells increases the length of time series the LSTM is able to process;
three gates (input, output and forget gates) regulate the flow of information into and out of the memory
cell, determining which information from the past is to be retained and which can be forgotten. In this
way, each member of the LSTM output becomes a function of the relevant input at previous timesteps.
The LSTM network consists of an input layer, one or more hidden layers, and an output layer. The layers
are connected by a set of updatable weights, with the same weights applying to all timesteps of the data.



Memory cells shadow each node on the hidden layer, retaining important information over long time
periods. Each node of the input layer represents a variable of the input data set. Observations are fed into
the network along with a pre-specified number of predictor values from previous timesteps (known as
the lookback length, or lag) which are cycled sequentially through the network. Network weights are
updated by backpropagating the gradient of the error between the modelled and observed outputs. For
detailed information on the mathematical functioning of the LSTM, see (Goodfellow et al., 2016) and
(Kratzert et al., 2018).
In this study, a separate LSTM is trained for each catchment. Input to the LSTMs are monthly averaged
measurements of: rainfall depth ($P$), potential evapotranspiration ($E$), average maximum daily
temperature over the month, and net monthly (effective) rainfall ($P^*$) computed for month $t$ by summing
daily effective rainfall, as shown here:

$$P_t^* = \sum_{d=0}^{d=days(t)} \max(0, P_d - E_d) \qquad 1$$

Standard scaling of the input data is performed per catchment as follows:

$$\tilde{X}_t = \frac{X_t - \mu_x}{\sigma_x} \qquad 2$$

where $X_t$ is an input variable for month $t$, $\mu_x$ is its mean and $\sigma_x$ its standard deviation over the training
period. The target variable for LSTM training is monthly average runoff. Observed runoff values are
scaled by taking the square root and then transforming to the range [-1,1] per catchment, as follows:

$$Y_t = 2\frac{\sqrt{Q_t} - Y_0}{Y_1 - Y_0} - 1 \qquad 3$$

where $Q_t$ is the observed runoff for month $t$, and $Y_0$ and $Y_1$ are the minimum and maximum square root
transformed flow over the training period, respectively. The square root transform is chosen to be
conceptually consistent with the objective function of the WAPABA model calibration (as described
below, mean absolute error of the square roots of flows). Note that the same scaling constants
($\mu_x, \sigma_x, Y_0, Y_1$) used during LSTM training are also applied to LSTM inputs and targets for the testing
period. Using scaling constants only derived from the training data ensures that the training process is
not incorporating any information from the testing data set.
The loss function used for training the LSTM is the mean absolute error (MAE) performed on the
transformed runoff, as follows:





$$L = \sum_t |Y_t - \hat{Y}_t| \qquad\qquad 4$$

where $\hat{Y}_t$ is the output of the network for month $t$ and $Y_t$ is the transformed runoff for the same month.
Hyperparameters, or parameters controlling the LSTM training algorithm, were selected after a grid
search on a randomly selected catchment (14207) with a good length data record and tested on a small
additional subset of catchments. The hyperparameter space searched was: initial learning rate $\delta_0$ (1e-3
to 1e-4), sequence (lookback or lag) length (6, 9, 12, 15, 18, 21, 24 months) and number of hidden nodes
(10, 20, 30, 40, 50, 60). The hyperparameter set that performed the best predictions over the training
period selected for use in all LSTMs: 10 nodes on a single hidden layer, run with a sequence length 6
months, and an initial learning rate $\delta_0$ of 0.0001. Subsequent to this hyperparameter search on one
catchment, we investigated on all catchments the effect of raising the initial learning rate for faster
convergence while using input and recurrent dropout to prevent overfitting. Empirically, and counter to
our intuition, this never improved training performance so an initial learning rate $\delta_0$ of 0.0001 was kept.
The learning rate was allowed to vary during training with a patience of 3 epochs without improvement
before multiplying by a factor of 0.2 to obtain a new learning rate. The dataset was divided into 400
steps-per-epoch for training; data was sent through the model in batches with a weight update after each
(an epoch, or iteration, is concluded when the entire dataset has been run through the model once). The
LSTM training was implemented using a gradient descent algorithm run for a maximum of 100 epochs.
Training was set to stop early if the training error failed to decrease over 5 consecutive epochs. The
LSTMs were implemented with Tensorflow in Python. The code was designed to use numeric seeds to
have reproducible outcomes, which is often not the default behavior of many components of Tensorflow
or other deep learning frameworks.
**2.2.2. WAPABA rainfall-runoff models**
The WAPABA model is a conceptual monthly rainfall-runoff model introduced by Wang et al. (2011).
The model is an evolution of the Budyko framework proposed by Zhang et al. (2008) where water fluxes
are partitioned using parameterised curves. The model uses two inputs, mean monthly rainfall and
potential evapotranspiration, and operates in five stages. First, input rainfall is split between effective
rainfall that will eventually leave the catchment, and catchment consumption that replenishes soil
moisture and evaporates. Second, catchment consumption is portioned between soil moisture
replenishment and actual evapotranspiration. Third, effective rainfall is partitioned between surface
water (fast) and groundwater (slow) stores. Fourth, the groundwater store is drained to provide a
baseflow contribution. Fifth, the surface water and baseflow are added to obtain the final simulated
runoff for the month. The model has five parameters described in Table 2.





Table 2 WAPABA model parameters

| Name | Description | Unit | Minimum | Maximum |
|------|-------------|------|---------|---------|
| **alpha1** | Exponent of the catchment consumption/effective rainfall curve | Dimensionless | 1.0 | 10.0 |
| **alpha2** | Exponent of the soil moisture storage/evapotranspiration curve | Dimensionless | 1.0 | 10.0 |
| **Beta** | Partition between groundwater recharge and surface runoff | Dimensionless | 0.0 | 1.0 |
| **Smax** | Maximum water-holding capacity of soil store | mm | 5.0 | 6000.0 |
| **Inverse K** | Inverse of groundwater store time constant | 1/day | 0.000274 | 1.0 |


A separate WAPABA model is run for each study catchment. The WAPABA models were trained
(calibrated) and tested (validated) over the same periods as the LSTMs: 1950 to 1995 inclusive for
training, and 1996 to June 2020 for testing. WAPABA parameters were optimized over the training
period using the Shuffle Complex Evolution algorithm (Duan et al., 1993) with the Swift software
package (Perraud et al., 2015). The objective function used for the WAPABA models is the same as the
one used for LSTM, i.e. the mean absolute error (MAE) on the square root of runoff (see Equation 4).
**2.3.    Performance evaluation**
Predictions from the conceptual (WAPABA) and machine learning (LSTM) models for all catchments
are compared to observed runoff, assessing each models' predictive capabilities on the set of catchments.
Runoff prediction performance is reported here using the following metrics.
The Nash Sutcliffe Efficiency (NSE, (Nash and Sutcliffe, 1970)) is the most often used performance
metric in hydrology. It can be considered a normalised form of mean squared error (MSE) and is defined
as:

$$NSE = 1 - \frac{\sum_t (Q_{obs}^t - Q_{mod}^t)^2}{\sum_t (Q_{obs}^t - \mu_{obs})^2} = 1 - \frac{E}{V}$$

where $Q_{obs}^t$ and $Q_{mod}^t$ are the observed and modelled discharges for month $t$, respectively, and $\mu_{obs}$ is
the average observed discharge over the training or testing period. The ratio of the sum of squared errors,
$E = \sum_t (Q_{obs}^t - Q_{mod}^t)^2$, to the variance, $V = \sum_t (Q_{obs}^t - \mu_{obs})^2$, is subtracted from a maximum score of 1. An
NSE closer to 1 indicates better predictive capability of the model, and an NSE less than 0 indicates the
model mean squared error is larger than the observation variance.





The NSE metric alone cannot provide an accurate description of model performance due to its focus on
high flow regime (Schaefli and Gupta, 2007). The reciprocal NSE focuses the error metric on low flows
(Pushpalatha et al., 2012) by comparing the reciprocals of the observed and modelled flows. It is
calculated as:

$$RecipNSE = 1 - \frac{\sum_t \left( \frac{1}{(Q_{obs}^t + 1)} - \frac{1}{(Q_{mod}^t + 1)} \right)^2}{\sum_t \left( \frac{1}{(Q_{obs}^t + 1)} - \frac{1}{(\mu_{obs} + 1)} \right)^2} \qquad 6$$

The Kling-Gupta efficiency (KGE, (Gupta et al., 2009)) provides an alternative to metrics based on sum
of squared error such as the two previous ones, by equally weighting measures of bias of the mean,
variability, and correlation into a single metric as follows:

$$KGE = 1 - \sqrt{\left(1 - \frac{\mu_{sim}}{\mu_{obs}}\right)^2 + \left(1 - \frac{\sigma_{sim}}{\sigma_{obs}}\right)^2 + (1 - \rho)^2} \qquad 7$$

where $\mu_X$ and $\sigma_X$ are the mean and the standard deviation and $\rho$ is the Pearson correlation coefficient
between the simulated and observed data.
Finally, bias is a measure of consistent under-forecasting or over-forecasting of the mean, defined as:

$$Bias = \frac{\mu_{sim} - \mu_{obs}}{\mu_{obs}}. \qquad 8$$

**Comparison of performance metrics between catchments using normalised indexes**
When comparing metrics across model types and catchments, a normalised difference in NSE values is
used. The NSE metric can reach into large negative values in dry catchments when the variance of the
observations is very small compared to the model errors (Mathevet et al., 2006), as can be seen from
Equation 5. Differences between large negative values of NSE have a much smaller implication than the
same absolute difference between values of NSE closer to 1. To allow for a comparison between the
WAPABA and LSTM models at catchments of various aridities, the normalised difference in NSE is
calculated following Lerat et al. (2012):

$$Diff\_NSE_{norm} = \frac{NSE_2 - NSE_1}{(1 - NSE_1) + (1 - NSE_2)} = \frac{NSE_2 - NSE_1}{2 - (NSE_1 + NSE_2)} \qquad 9$$

where $NSE_1$ and $NSE_2$ are the NSE values corresponding to the two models to be compared. Substituting
in $NSE = 1 - \frac{E}{V}$ from Equation 5 into Equation 9, the normalised difference in NSE can be seen to
represent a percentage difference in the sum of squared errors between the two models being compared:





$$Diff\_NSE_{norm} = \frac{NSE_2 - NSE_1}{2 - (NSE_1 + NSE_2)} = \frac{E_1 - E_2}{E_1 + E_2} \qquad 10$$

A similar formula is applied to reciprocial NSE and KGE. The normalised difference between the bias
for two models is calculated as:

$$Diff\_Bias_{norm} = \frac{|Bias_1| - |Bias_2|}{|Bias_1| + |Bias_2|} \qquad 11$$

To simplify the comparison of model results across the large number of catchments, model performances
at each catchment are classified as similar if the normalised difference between WAPABA and LSTM
metrics lies within +/- 0.05 at that catchment, following Lerat et al. (2020). Therefore in this paper, a
'similar' NSE denotes that the sum of squared errors of the WAPABA and LSTM models at an
individual catchment differ by no more than 5%. For differences greater than this, the catchments are
classified by the model type producing the higher metric. The selection of the threshold of 0.05 was
based on the recommendations of (Lerat et al., 2020) and the authors' experience relative to the use of
the NSE, KGE and bias metrics.

## 3. Results

For each of the study catchments, a WAPABA model and an LSTM model have been trained using
monthly data over the training period, and the prediction performance of the models are evaluated here
on monthly data from the testing period (data unseen by the model during training) using the metrics
described above. A general comparison of WAPABA and LSTM prediction performance is first made
over all catchments with a continental-scale analysis of the performance metrics, to determine:
1) the proportion of overall catchments for which the WAPABAs or the LSTMs produced

328        better predictions, and

2) differences at individual catchments in WAPABA versus LSTM prediction performance.
A comparison of model performance is then made in relation to various catchment and time series
characteristics (eg. catchment size, flow level, record length), to determine if an association exists
between these properties and the relative performance of the conceptual and machine learning models.
**Example prediction results**
As a sample of the modelling output, Figure 2 shows the WAPABA and LSTM runoff predictions along
with the corresponding observed runoff for the three stations highlighted in Figure 1 (over the testing
period). These hydrographs are representative of a wet catchment in Northern Queensland (Mulgrave
River at the Fisheries, 111005), a temperate catchment in NSW (Mann River at Mitchell, 204014), and
a dry, intermittent catchment in Western Australia (Blackwood River at Winnejup, 609012). NSE values
of each of the predictions are noted. The WAPABA and LSTM predictions both match the observed
data reasonably well in the three catchments. The performance of the models, in particular for the
Blackwood River at Winnejup is remarkable because of the difficulty in modelling dry, intermittent
catchments (Wang et al., 2020). The next sections provide a more detailed assessment of the performance
over all catchments using quantitative metrics.

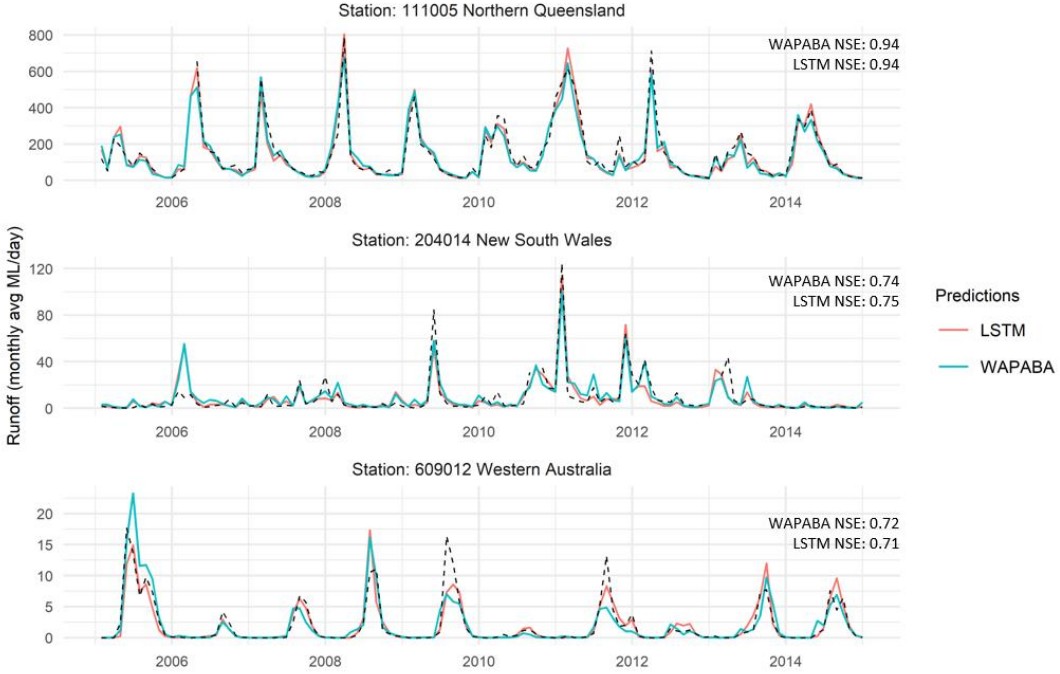


Figure 2: Observed data (black dashed line) and predicted runoff (by WAPABA and LSTM models) over the testing period
for the Mulgrave River at the Fisheries (111005), Mann River at Mitchell (204014) and the Blackwood River at Winnejup
(609012). Catchment locations are shown on Figure 1.
**Large-sample performance summary**
The general runoff prediction performance of WAPABA and LSTM models on a continent-wide basis
is summarized in Figure 3. From the models run for each catchment, metrics are determined on the
training portion (calibration) and testing portion (validation) separately and gathered here in boxplots.
Median and quartiles of NSE, reciprocal NSE, KGE and Bias over all catchments are shown for each
model type, with each data point representing an individual catchment. All data is shown on the top
panel, and due to a few large (negative) outliers the same figure is shown with a restricted y-axis for
visualization purposes on the lower panel. Higher values of the first three metrics (NSE, reciprocal NSE





and KGE) indicate a better match of predicted runoff with observed runoff, whereas lower values of
Bias indicate better prediction results.
Figure 3 shows that across the set of study catchments the median values of NSE, Reciprocal NSE, and
KGE are slightly higher for LSTM than for WAPABA during both the training and testing phases. Bias
has a slightly lower median for the LSTM. As expected, both model types perform better during the
training phase than the testing phase for all metrics. The interquartile ranges increase from training to
testing (longer boxes during testing), indicating a greater spread of performance results when the models
are run on data not seen during the training phase. Over all catchments, the median NSE is: 0.74 with
the WAPABA models and 0.76 with the LSTM models (on testing data). See Table 3 for median values
of all metrics.

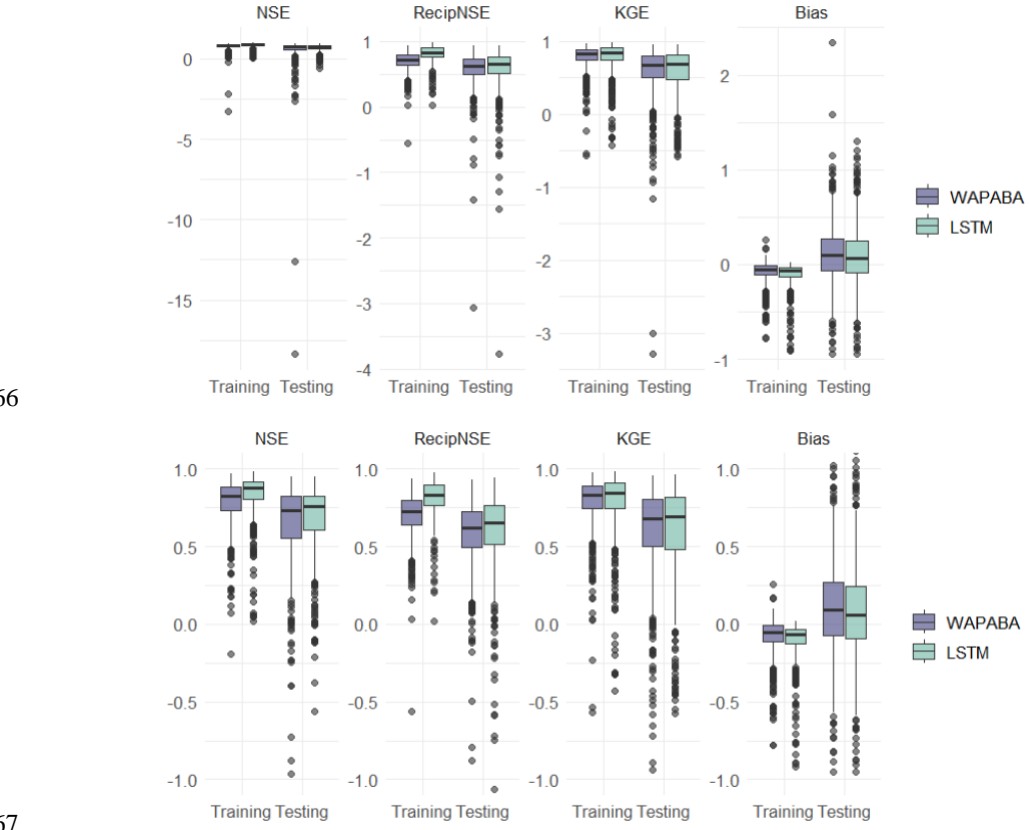


Figure 3: Performance metrics summary for the set of 496 catchments (zoomed in on lower panel, excluding outliers < -1).
Median values of LSTM performance metrics are slightly higher than WAPABA for NSE, Reciprocal NSE and KGE, and
slightly lower for Bias (lower Bias is preferable). For all four metrics on both models, the training results were better than
the testing results, with the longer testing boxes indicating more spread in performance results when predicting on new
data.



Table 3: Median values of metrics over the set of catchments (n=496)

|  | WAPABA | LSTM |
| --- | --- | --- |
| **NSE** | 0.74 | 0.76 |
| **Reciprocal NSE** | 0.62 | 0.65 |
| **KGE** | 0.68 | 0.70 |
| **Bias** | 0.09 | 0.06 |


Aggregated performance metrics may mask performance variability within certain aspects of the time
series (Mathevet et al., 2020). The KGE has the benefit of being easily decomposed into three
components for further error analysis: bias of the mean (ratio of mean of simulations to mean of
observations), bias of variability (ratio of standard deviation of simulations to standard deviation of the
observations), and correlation (matching of the timing and shape of the time series to the
observations).
In Figure 4, model performance is assessed with respect to each component of the KGE metric.
Boxplots of the decomposed KGE components are shown by model type and training/testing period.
During testing, the medians of bias of the mean and standard deviation are above zero and greater for
WAPABA than LSTM. This indicates that mean streamflow and streamflow variability tend to be
overestimated more by the WAPABA models compared to the LSTMs. With the LSTM, streamflow
variability is more prone to underestimation (median below zero). For bias of the mean and standard
deviation, the depth of the boxplots increases from training to testing, indicating the bias values from
individual catchments are more diverse during the testing period.
The scatterplots in the lower part of Figure 4 compare the KGE components at individual catchments
for the WAPABA and LSTM models (each dot represents a catchment), separately for training and
testing portions of the data. Most values of bias of the mean (left column) are between 0 and 1 during
training (underestimating) yet during testing values extend beyond 2, indicating the mean flow in
many catchments is overestimated by both model types on the testing data. The observable correlation
in testing period bias of the mean between WAPABA and LSTM indicates that this error is not





specific to model type. Correlation between simulations and observed data is similar for both model
types and remains relatively constant between training and testing (right column).

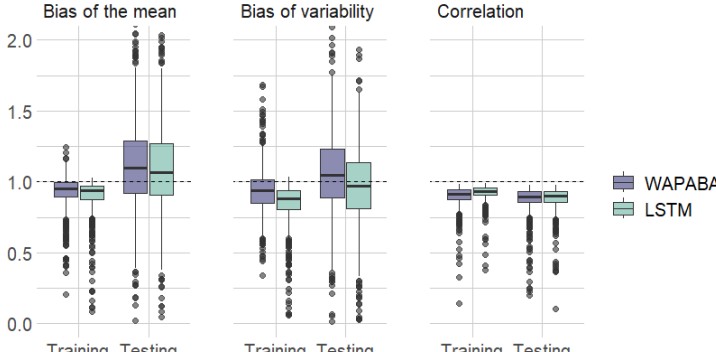


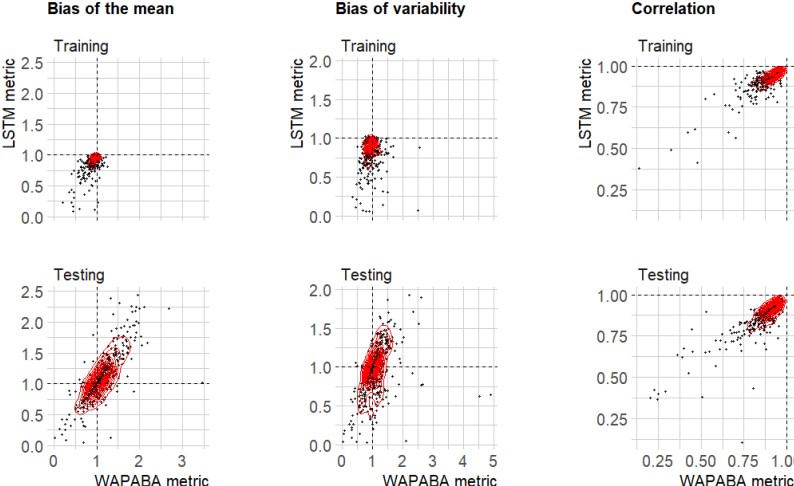


Figure 4: KGE decomposition into three components: bias of the mean, bias of variability, and correlation. Each dot
represents an individual catchment (large outliers have been omitted for visualization purposes.) The mean flow and
variability (left and middle columns) tend to be underestimated during training and both under- and overestimated during
testing by both model types. The correlation (right column) remains similar during training and testing.

**Performance differences at individual catchments**
The differences between WAPABA and LSTM performance at each catchment (eg. $NSE_i =$
$NSE_{i,WAPABA} - NSE_{i,LSTM}$ for catchment $i$) are summarised in Figure 5. Values above zero indicate
higher metrics obtained by WAPABA, and values below zero indicate higher metrics obtained by the
LSTM model at a specific catchment.





The boxplots indicate that median differences in WAPABA and LSTM prediction performance at each
catchment (measured by NSE, Reciprocal NSE, KGE and Bias on the testing data) are very close to zero.
However, there are outliers (black dots) representing large performance differences between WAPABA
and LSTM models, both positive and negative. These indicate that each model provides advantages for
predicting runoff in certain catchments. In this figure the boxplots are restricted to the range [-1,1] for
visualisation purposes. A version of this figure including the large outliers is provided in Figure A1 of
the Appendix.

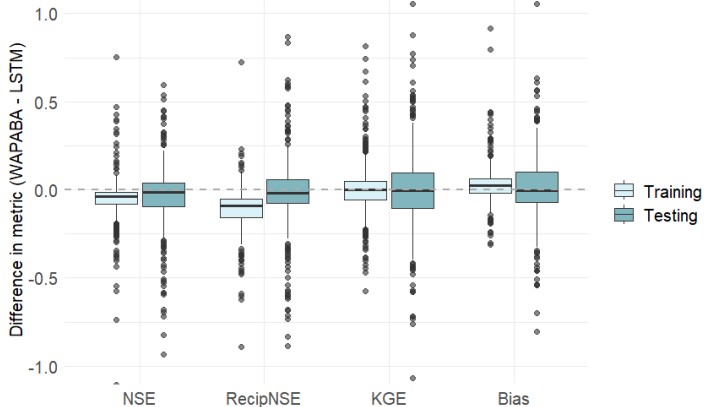


Figure 5: Difference in the metrics (WAPABA – LSTM) for each catchment. A positive value indicates WAPABA has a
higher metric for that catchment, and a negative value indicates LSTM has a higher metric. The median difference in each
metric lies close to zero for the testing portion of the dataset, signifying overall similarity in catchment-specific metrics
between model types. Large negative outliers have been excluded from this figure for visualisation purposes, but are
included in the reproduction in the Appendix.

This data set represents a range of catchments across Australia, some being characterised by highly arid
conditions. To enable comparisons between these diverse catchments, the impact of large negative NSE
values which can occur at very dry catchments is minimised by calculating the normalised differences
in NSE between the WAPABA and LSTM predictions at each catchment, as per Equation 9. The
normalised differences fall into the range [-1,1], facilitating comparison. This distribution is shown in
Figure 6 for the 496 catchments. The portion of the distribution lying to the right of the vertical dashed
line corresponds to catchments with better prediction by WAPABA and catchments to the left have
better prediction by LSTM. The x-axis corresponds to percentage differences between the sum of
squared errors of the two model types (ie. -0.5 indicates a 50% performance gain by LSTM and 0.5
indicates a 50% performance gain by WAPABA).



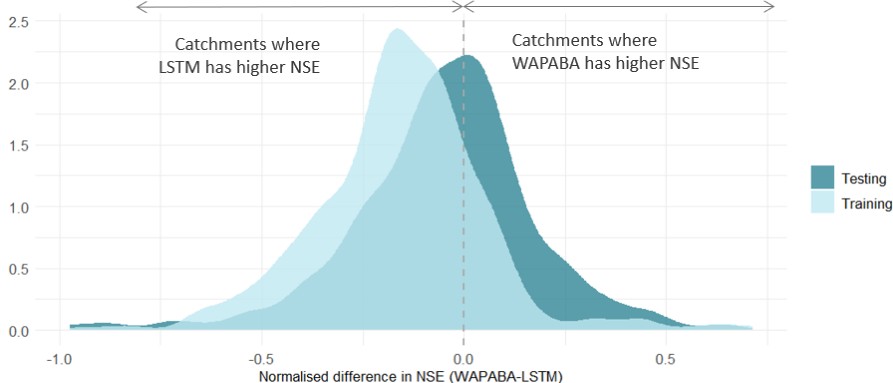


Figure 6: Distribution of normalized differences between WAPABA and LSTM prediction performance at individual catchments (measured by NSE). The values on the x-axis represent percentage/100 difference in sum of squared errors between WAPABA and LSTM at the same catchment (ie 0.5 –> 50% difference in sum of squared errors). The catchments under the curve on the right of the dashed line have better predictions by the WAPABA model and on the left by the LSTM model.


In Figure 6, we see that during the training period the majority of catchments are to the left of the line
indicating better prediction by LSTM, and in the testing period there is a more even split. The median
normalised difference in NSE across the 496 catchments over the training period is -0.15 (mean -0.16)
and -0.04 (mean -0.05) during the testing period. This equates to a median 15% performance advantage
by LSTM versus WAPABA during training and 4% during testing based on sum of squared errors.
This figure suggests that in general there is little overall advantage of either the WAPABA or LSTM
models when predicting on unseen data across the whole sample of catchments. However, the width of
the distribution indicates that both the WAPABA and LSTM models have advantages at certain
individual catchments, which will be explored in the next section.
Figure 7 quantifies the proportion of catchments with similar or better prediction performance by either
WAPABA or LSTM (on the testing data). 'Similarity' is defined here as an absolute normalised
difference in NSE of less than 0.05 between WAPABA and LSTM predictions, meaning the sum of
squared errors of the WAPABA and LSTM models at an individual catchment differ by no more than

454 5%.

The LSTM models produce similar or higher NSE values for 69% of the catchments when tested on data
not seen during the training process (and 89% of the catchments during training, not shown). It can also
be seen that 70% of catchments have similar or higher reciprocal NSE (focusing on low flow predictions)
with LSTM , 61% have similar or higher KGE with LSTM, and 57% have similar or lower Bias with
LSTM model compared to WAPABA on the same catchment.




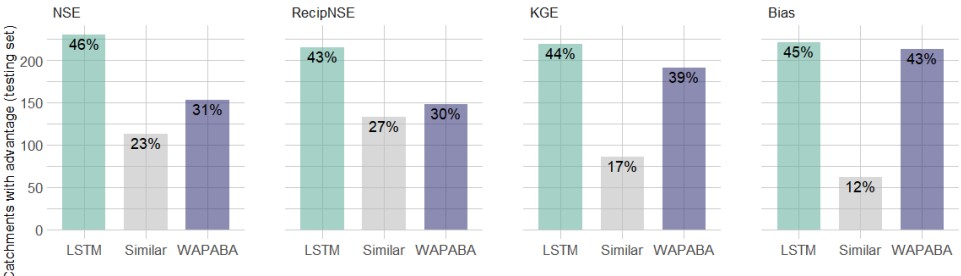


Figure 7: Percentage of catchments with similar or better performance metrics on the testing portion of the data (note better
Bias is lower, all others is higher). For catchments in the 'similar' category, the sum of squared errors of the WAPABA and
LSTM predictions differ by less than 5%. The LSTM model produces predictions with similar or higher NSE values
compared to the WAPABA predictions for 69% of the catchments.

**468 Prediction performance comparison by catchment or time series characteristics**

In this section, we investigate if the abilities of WAPABA and LSTM to accurately predict runoff at
individual catchments vary based on attributes such as catchment area, flow level and length of historical
record.

*472 Catchment size*

Figure 8 shows the association of prediction performance with catchment area. The left panel shows the
catchment area compared to the normalised difference in NSE between LSTM and WAPABA prediction
performance for each catchment. Data points are coloured according to the model that produced the
better prediction for that catchment. This figure indicates the largest performance gains of LSTM versus
WAPABA occurred in large catchments (points furthest to the left are found in the upper portion of
figure). Splitting the catchments into quintiles by area, we can analyse the results for the largest 20% of
catchments. Of these catchments, over three-quarters (78%) had similar or better runoff predictions with
the LSTM (with similarity defined as less than 5% difference in sum of squared errors compared to
WAPABA predictions). In this top quintile of catchments, those with higher NSE values from the LSTM
show a greater average advantage (average 24% lower sum of squared errors, maximum 97% lower),
than those with better WAPABA predictions (average 15% lower sum of squared errors, maximum 65%
lower).
The mirrored histogram in the right panel of Figure 8 shows catchments stratified into bins by area (log
base 10), coloured and counted by the model type that produced the better runoff prediction at each
catchment. The LSTM models produced higher NSEs for a greater number of catchments than the
WAPABA models in all of the bins, except the lowest bin (where n=1).





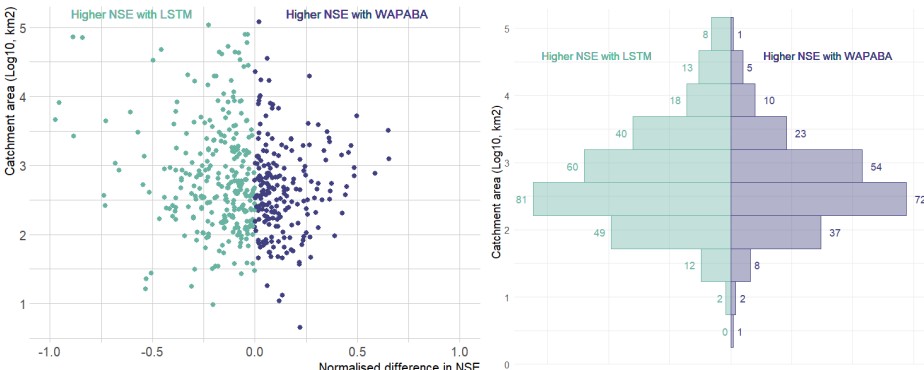

Figure 8: Model performance by catchment size. Left panel: Each data point represents the normalized difference in prediction performance at an individual catchment, arranged by catchment size. The spread of data points in the top left quadrant indicates that in large catchments the performance gain of LSTM versus WAPABA can exceed 90% in terms of sum of squared errors. Right panel: count of catchments in each size bin that have better performance with each model.

***Flow level***

Model performance is compared for high, medium and low flow portions of the time series. For each station, each observation is categorised based on its flow level. High flows are defined here as the top 5% of flow values and low flows as the lower 10% of flows at each station (calculated excluding zeros) over all observed data during the study period. The training and testing portions of the time series over all the catchments have different distributions of flow levels, as listed in Table 4. During the testing portion of the study period, conditions are dryer with more no-flow and low-flow observations, and fewer medium- and high-flow observations than during training.

Table 4: Distribution of flow levels during training and testing

| Flow level | Training observations (n) | Testing observations (n) |
|---|---|---|
| No flow | 18,728 | **21,690** |
| Low | 11,967 | **14,668** |
| Medium | **127,584** | 96,089 |
| High | **9,192** | 4,203 |

For comparison purposes, both observed and modelled flows are standardised by station based on the mean and standard deviation of all observations at that station during the study period. The observed mean is subtracted from each value before dividing by the standard deviation of the observations.





Figure 9 shows that when NSE is calculated separately for the low, medium and high flow measurements
at each catchment, both model types have similar NSE distributions. Medium flows are better predicted
(NSE peak closer to 1) than high flows, and low flows appear to be poorly represented by both
WAPABA and the LSTM.

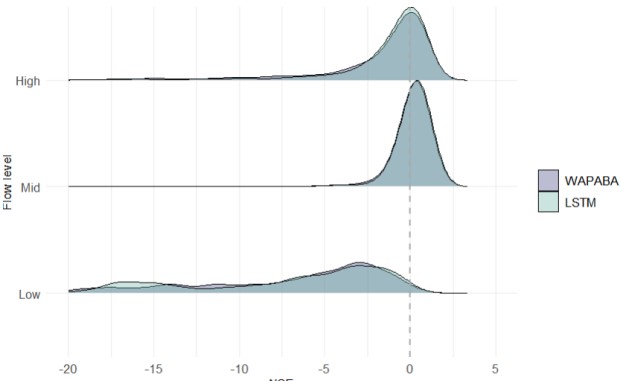


Figure 9: NSE distributions calculated separately by flow level over all catchments. Both model types have similar
distributions of NSE by flow. Medium flows are best represented, followed by high and then low flows.

Figure 10 compares the scaled modelled flow to the scaled observations for all testing observations at
all stations. Kernel density contours split the data into 10 density regions on each plot and a 1:1 line is
added to aid interpretation. The lower panel focuses on the regions of highest density for each subset of
flows. For no-flows and low flows (left two panels), the densest portions of the observation/prediction
clouds are closely aligned along the 1:1 line for both WAPABA and LSTM. The magnitude of the
outliers (beyond the outermost contour) is greatest above the 1:1 line indicating that prediction errors
for no-flows and low flows are dominated by overestimations. For medium flow levels, the contours
again follow the 1:1 line. The contours tend to expand upwards as flow size increases, indicating a
tendency towards more overestimation with higher flows. The shape of the contours is similar for both
models. On the upper panel it can be seen that the edges of the data cloud also expand upwards and
outwards as the flows increase. The medium flow prediction errors with largest magnitude tend to be
overestimations, with the WAPABAs producing greater overestimations than the LSTMs on the higher
flows (still in this medium-flow subset). For high flows (on the far right panel), the majority tend to be
underestimated by both LSTM and WAPABA (central density located below the 1:1 line), though there
is a difference in the outliers – most of the larger errors in LSTM high flow predictions are
underestimations, whereas the high-magnitude WAPABA errors are both over- and underestimations of
high flows.

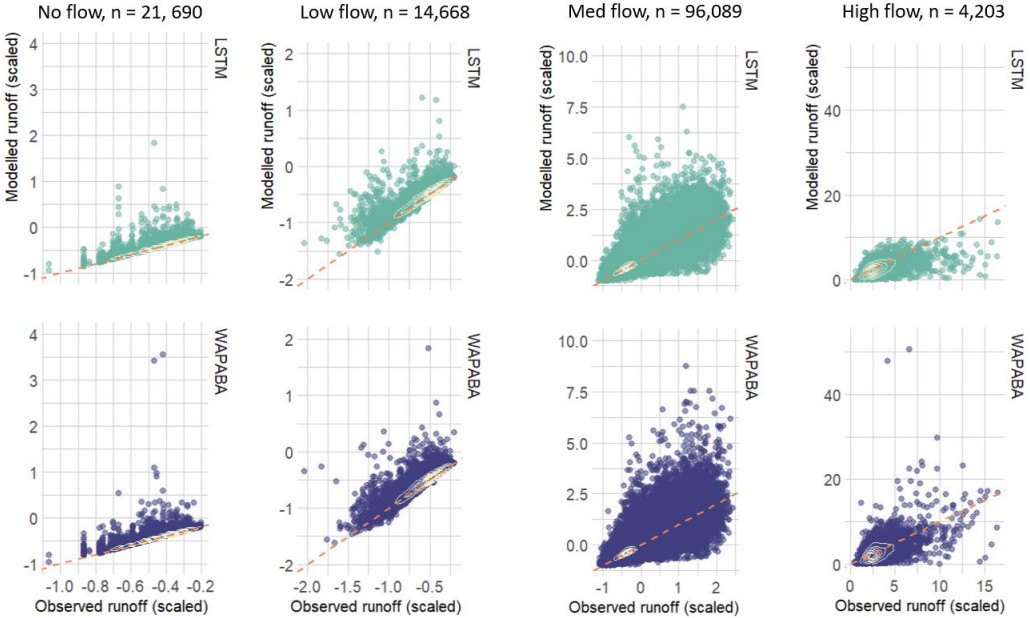



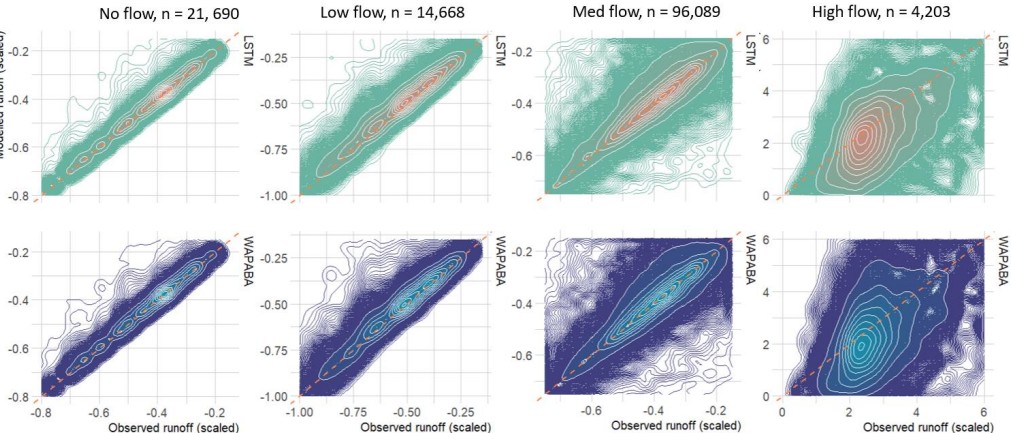


Figure 10: Prediction performance related to flow level. Upper panel: Observed vs. modelled flow pairs at all stations, separated into no-flow, low, medium and high flows [testing data only]. Densest portion of the data cloud is identified with density contours. Data are standardized based on observed mean and standard deviation. Lower panel: Comparison of density distributions of the data, zoomed in on the kernel density contours. In general, the largest errors on medium flows tend to be overestimations (by both models) and on high flows tend to be underestimations (by WAPABA and LSTM) or overestimations (by WAPABA).

### *Poorly predicted catchments*

Figure 11 compares the NSEs for WAPABA and LSTM runoff predictions by catchment. Each dot
represents an individual catchment, coloured according to the model with higher NSE at that catchment.



The top left quadrant contains catchments where $NSE_{WAPABA} < 0$ and $NSE_{LSTM} > 0$ (n=19), and the lower
right quadrant contains catchments where $NSE_{LSTM} < 0$ and $NSE_{WAPABA} > 0$ (n=5).

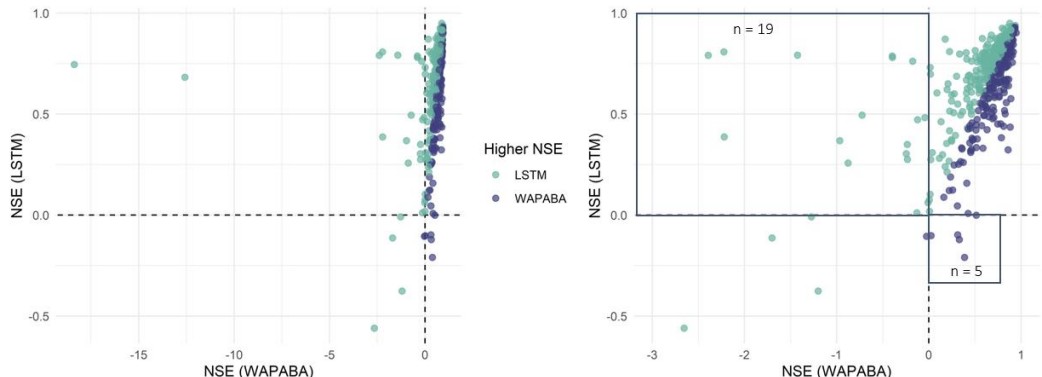

Figure 11: Comparison of NSEs on testing data - each data point represents the WAPABA and LSTM values of NSE for a
single catchment, coloured by the model which provides the best prediction at that catchment. On the right panel, two far-
left outliers have been removed to enable better viewing of the other datapoints. Catchments in the upper left quadrant are
those in which runoff is poorly predicted by WAPABA (NSE < 0) and better predictions (NSE > 0) are obtained with
LSTM. The lower right quadrant correspondingly shows catchments in which the NSE values from LSTM are below 0 and
WAPABA has better predictions (NSE > 0).

WAPABA and LSTM predictions at each catchment are classified into poor (NSE < 0), fair (0 <= NSE
<= 0.5) or good (NSE > 0.5) categories. In this set of catchments, the runoff at 5 catchments is poorly
predicted (NSE < 0) by both model types (lower left quadrant of Figure 11). All other catchments are
better represented by one model or the other, with either WAPABA or LSTM producing predictions
with NSEs above 0.
For the 5% (n=24) of overall catchments that are poorly represented by WAPABA (NSE < 0), runoff
predictions at 23 of these catchments (96%) are improved with use of the LSTM. In fact, one-third (n=8)
of these have 'good' predictions by LSTM (NSE > 0.5). Conversely, for the 2% of catchments (n=10)
that are poorly represented by LSTM, 60% are improved with use of WAPABA, and one-tenth (n=1)
have 'good' predictions by WAPABA. Figure 12 depicts the number of catchments poorly represented
by each model and how these specific catchments are represented by the alternate model. For half of the
catchments with poor LSTM predictions, WAPABA does poorly as well; whereas in 79% of the
catchments with poor WABAPA predictions, fair or good predictions were obtained with the LSTM.





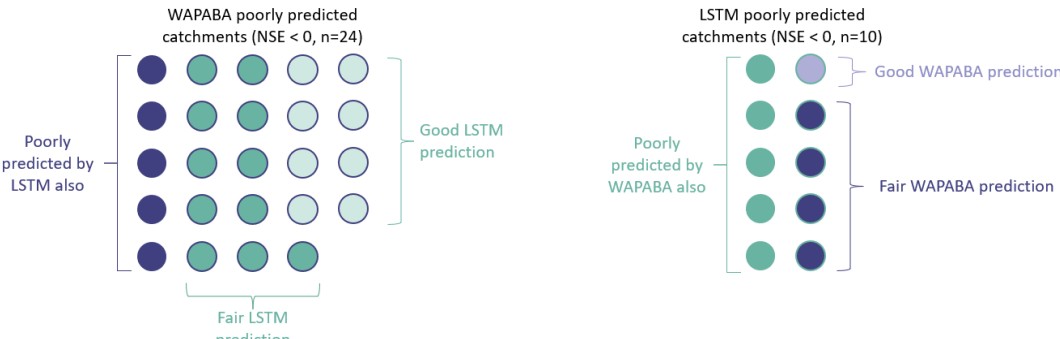


Figure 12: Number of catchments with poor runoff predictions by each model type. Colouring indicates the prediction
results from the alternate model type. One-third of WAPABA poorly predicted catchments have good predictions with the
LSTM. One-tenth of LSTM poorly predicted catchments have good predictions with the WAPABA. Results are denoted as
poor (NSE < 0), fair (0 <= NSE <= 0.5), or good (NSE > 0.5).

*Generalising to changing conditions*


The ability of a model to generalise outside of the conditions encountered during training is important,
especially in the context of a changing climate. A model that is able to make predictions on unseen
(testing) data to a comparable performance level as on the training data will provide confidence in
making predictions into the future when external conditions are not expected to remain constant. In this
data set we know that conditions differ between the training and testing data, with wetter climate
conditions during the training period and a dryer testing period.
It was found that 2% (n=11) of WAPABA models struggled with generalising outside of the training
period, with 'good' (NSE > 0.5) runoff predictions during training but 'very poor' predictions (NSE < -
0.5) during the testing period. The testing predictions for all of these catchments were improved by use
of the LSTM, and at 4 of these catchments 'good' predictions (NSE > 0.5) were obtained with the
LSTMs. Conversely, one LSTM model produced 'good' training runoff predictions and 'very poor'
testing predictions. This catchment was one of the 11 that also had poor generalisation (and 'very poor'
predictions) with the WAPABA.

*Historical record length and data set size*


The performance of each model type is compared to the length of historical records available at each
station. Training data length has been categorized here as 14-25 years (38% of stations), 25-35 years
(40%), and 35-47 years (23%).
Figure 13 (top panel) shows prediction performance varying slightly with record length (for visualisation
purposes, this figure is shown without large negative outliers – the figure including outliers is provided




in Figure A2 of the Appendix). Stations with medium record length tend to have slightly better
predictions according to the four metrics than those with shorter records. The performance levels tend
to even out as record lengths increase beyond 35 years, and there is even a slight decline in the WAPABA
reciprocal NSE.
Considering catchments individually, the median normalised difference in NSE between WAPABA and
LSTM predictions (on testing data) is just slightly below zero for all record lengths: -0.03 (<25 years of
record), -0.04 (25-35 years), -0.04 (>35 years). This indicates that, in each of the short, medium and long
record length categories, at least half of the individual catchments have higher NSEs with the LSTMs.

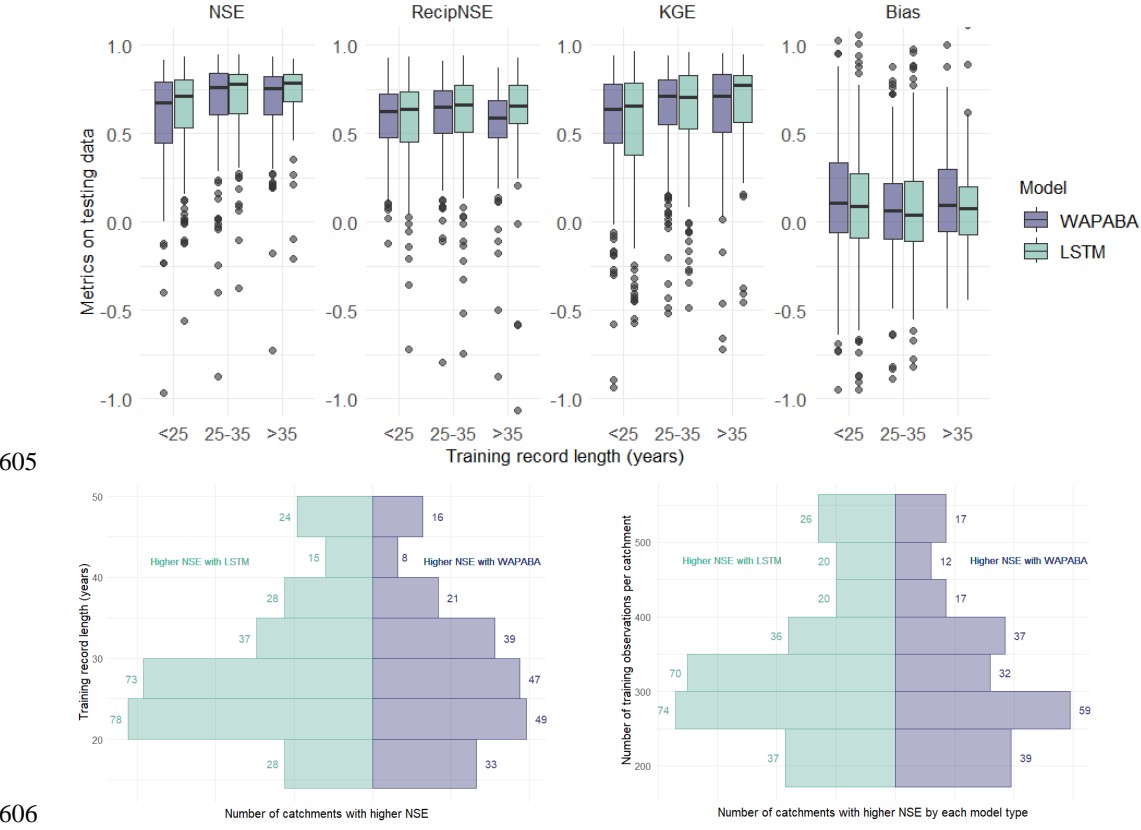


Figure 13: Effect of record length and training data size on prediction performance for each model type. Upper panel: Medians
of the NSE and KGE on testing data increase with record length for both WAPABA and LSTM predictions (large negative
outliers have been excluded for visualization purposes, but are included in the corresponding figure in the Appendix). Lower
left panel: Advantage of each model in 5-year increments of record length based on NSE values. Lower right panel:
Advantage of each model based on number of training observations.



The mirrored histogram in the lower left panel of Figure 13 quantifies the number of catchments within
5-year bins of record length in which runoff is better predicted by the LSTMs or by the WAPABAs. In
six of the eight bins, the majority of catchments are better represented by the LSTMs.
Comparing performance based on the number of years of record does not take into account the actual
size of the data sets, since measurement frequency differs at each station. Catchments in this study have
between 172 and 564 training data observations (425-846 including testing data). The lower right panel
of **Error! Reference source not found.** shows the number of catchments best modelled by the W
APABA or LSTM model (determined by higher NSE on the testing data) in relation to the number of
training observations. Median NSE values of both the WAPABA and LSTM predictions increased with
increasing number of training data points (not shown). Of particular note is that runoff at catchments
with the smallest data sets (less than 250 training data points) were similarly well predicted by both
LSTM (median NSE = 0.67) and WAPABA (median NSE = 0.66).

## 4. Discussion

The machine learning models were found to match the conceptual model performance for the majority
of catchments in this study. When considered over the entire catchment set, the median NSE of runoff
predictions was 0.74 with the WAPABA models and 0.76 with the LSTMs (on the testing data). The
medians of other metrics were similarly aligned.
When considering the differences between models in predicting runoff at individual catchments, LSTM
performance was similar to or exceeded WAPABA performance in 69% of the catchments in this study
(based on the NSE metric). The median differences in metrics (NSE, Reciprocal NSE, KGE and Bias)
between the model types at individual catchments were close to zero, though the range of differences
was wide in both directions suggesting many catchments had noticeable prediction advantages with
either the WAPABA or LSTM models.
Medium flows were similarly well represented by both model types, with less accurate predictions for
high flows and worse again for low flows. Both WAPABA and LSTM tend to overestimate low flows;
high flows are noticeably underestimated by LSTM and both over- and underestimated by WAPABA.
Across all flow levels, the mean flow is prevalently overestimated during testing for both model types,
though slightly more so by WAPABA (higher bias of the mean). This overestimation is expected as the
testing period in this study is drier than the training period and it is common to have an overestimation
of mean during dry periods (Vaze et al., 2010). Streamflow variability tends towards overestimation by
WAPABA and underestimation by LSTM.





Larger catchments were found to have the potential for greater prediction improvements with the LSTM.
This finding supports the work of (Fluet-Chouinard et al., 2022), who found that deep learning methods
compete especially well with traditional models in larger non-regulated rivers where the influence of
time lags is significant.
Though it is known that machine learning models generally benefit from large amounts of training data,
it is often not possible to provide large hydrological data sets. In this comparison, shorter training record
lengths did not affect one model type more than the other; the catchments with the smallest training data
sets (less than 250 observations) did not show a distinct prediction advantage with either WAPABA or
LSTM (median NSEs of 0.66 and 0.67 respectively).
In past studies, traditional models have been found to struggle to make accurate runoff predictions under
shifting meteorologic data (Saft et al., 2016). This is an issue that researchers have noted deep learning
models may have the potential to overcome (Li et al., 2021, Wi and Steinschneider, 2022). In this study,
the variation in differences in prediction performance at individual catchments is more evident during
the testing portion than the training portion of the time series, implying that the WAPABA and LSTM
models may each have advantages or drawbacks for generalising to unseen data on various catchments.
It was found that in catchments where the WAPABA models provide good runoff predictions during
training but struggle to make accurate predictions on new data, the LSTM provides improved predictions
in all cases (for those with testing NSE < 0 with WAPABA, all bar one had NSE > 0 with the LSTM).
In the opposite case, where the LSTM produced substantially poorer predictions on testing data than
training data, these predictions were not outdone by WAPABA. This improvement in predicting beyond
conditions experienced during training will become progressively important as climate change
continues.
Certain caveats are acknowledged regarding the metrics used here. It is possible that the use of individual
metrics to compare predictions along the entire length of the time series may mask any variability in
model performance that occurs in subperiods of the time series (Clark et al., 2021, Mathevet et al., 2020).
These limitations were addressed by comparing high, medium and low flow periods separately, though
there are many other subdivisions of the time series that we have not included in the scope of this study.
WAPABA is only one example of a conceptual rainfall-runoff model and there are others that could
have been chosen for this analysis, though fewer are suitable for comparisons at a monthly time step
than would be the case at the daily time step. Model comparisons in Wang et al. (2011), Bennett et al.
(2017) and the subsequent body of work with WAPABA in Australia have established WAPABA as a
reasonable benchmark against which to assess the machine learning model performance.





Future work may entail an expansion of the architecture and complexity of the LSTMs for modelling
this set of catchments, to determine what advantages could be gained from the use of more sophisticated
LSTMs. A simple LSTM has been used in this study, with a single layer and no catchment-specific
hyperparameter tuning. Through appropriate tuning of the models' architecture and hyperparameters for
each catchment, more accurate results could be expected. It is known that the performance of data-driven
runoff models is heavily dependent on the amount of lagged data that is used as input (Jin et al., 2022).
In this study, a lag of 6 months has been used for all of the catchments, based on a trial of up to 24
months lag on 10 random stations. As such, only temporal patterns of up to 6 months are captured by
the LSTMs used in this paper. Varying the length of lag on a catchment-specific basis may lead to better
performance.
Opportunities also exist for multiple time series analyses on this set of basins to capture patterns in
hydrologic behaviour that surpass the catchment scale. With multiple time series analysis we might
expect to see greater benefits in the use of machine learning over traditional hydrologic models, since
these large-scale studies present obstacles to traditional modelling due to their greater input data and
parameter requirements describing physical properties of the catchments (Nearing et al., 2021). This
may involve the development of hybrid models blending existing conceptual models with LSTMs, the
production of a global LSTM incorporating all time series, or transfer learning where a model is trained
on data from all catchments and then fine-tuned on a catchment-by-catchment basis, as in Kratzert et al.
(2019). Deep learning models have been found to produce better predictions when trained on multiple
rather than individual basins (Nearing et al., 2021), and it has been noted that the training of LSTMs on
large diverse sets of watersheds may help improve the realism of hydrologic projections under climate
change (Wi and Steinschneider, 2022).
The question of catchment-specific circumstances under which the LSTM may provide an advantage to
monthly rainfall-runoff modelling has been broached in an elementary fashion here, and a more
sophisticated investigation would be warranted in further studies. Investigation of multi-dimensional
patterns of catchment or climate characteristics that may be associated with differences in predictive
performance between the model types could lead to a greater understanding of the value that LSTMs
could add to hydrologic modelling.
Aside from scientific considerations, another important advantage of developing rainfall-runoff models
using a machine learning software framework is to easily share them among users and to benefit from
software optimisation provided by well-established frameworks such as Tensorflow, Keras, or Pytorch.
Better benchmark datasets and centralised repositories will be the key to advancement of machine
learning in hydrology (Nearing et al., 2021, Shen et al., 2021). Initiatives are being made to grow



reusable software for applying machine learning in hydrology and to benchmark these against other
approaches (Abbas et al., 2022) and (Kratzert et al., 2022).

### 5. Conclusion

A continental-scale comparison of conceptual and machine learning model predictions has been made
for monthly rainfall-runoff modelling on almost 500 diverse catchments across Australia. This large-
sample analysis of monthly-timescale models aggregates performance results over a variety of
catchment types, flow conditions, and hydrological record lengths.
The following conclusions have been found:
• The LSTM matches or exceeds the WAPABA prediction performance at a monthly scale for the
majority of catchments (69%) in this study.
• At individual catchments, the median difference in WAPABA and LSTM prediction
performance is close to zero but the distribution spreads in both directions, showing both model
types have advantages at certain catchments.
• At larger catchments, potential for a greater magnitude advantage of LSTM predictions over
WAPABA predictions was seen than at smaller catchments (though some large catchments were
better modelled by WAPABA).
• Both model types predict medium flows better than high or low flows. In general, the majority
of high flows were underestimated by both LSTM and WAPABA. However, whilst the largest
errors in high flow estimations by LSTM were underestimates, WAPABA also had some
tendency towards over-estimation of high flows. Therefore streamflow variability was found to
tend towards overestimation by WAPABA and underestimation by LSTM.
• More catchments are poorly predicted (NSE < 0) by WAPABA than by LSTM (5% vs. 2%). For
those poorly predicted by WAPABA, predictions at 96% were improved by use of LSTM. For
those poorly predicted by LSTM, 60% were improved by use of WAPABA.
• Generalisation is found to improve with use of the LSTM. At catchments in which WAPABA
produced good predictions on training data but very poor predictions on testing data, the testing
predictions were universally improved with use of the LSTM; the opposite case (poor
generalisation by LSTM improved by WAPABA) was not observed. In this data set, the testing
period was significantly drier than the training period. This has implications for making
predictions in the context of climate change.
• Training data set size has little affect on the models. Catchments with the smallest training data
sets (< 250 observations) were similarly well predicted by both model types.



With refinement of the LSTM model architecture and hyperparameter tuning specific to each catchment,
it may be possible to increase the proportion of catchments for which the LSTM provides good prediction
performance. Other benefits may be realised by combining multiple catchments within global models to
capture patterns that transcend catchment boundaries, or by transferring knowledge from data-rich
catchments to data-poor catchments, within Australia or from international source catchments.

**Author contributions**

PF and JMP designed the experiment with conceptual inputs from JL and SC. PF and JMP developed
the LSTM model code and performed the simulations, as JL performed the WAPABA simulations. SC
conducted the comparison and prepared the manuscript with contributions from all co-authors.

**Competing interests**

The authors declare that they have no conflict of interest.

**Acknowledgments**

The authors would like to thank the CSIRO Digital Water and Landscapes initiative for their support
and for the funding of this project.

**Data and code availability**

All data used in this paper are accessible through the website of the Australian Bureau of Meteorology.
Rainfall and potential evapotranspiration can be downloaded from the Australian Water Outlook portal
at the following address: https://awo.bom.gov.au/. Streamflow can be downloaded from the Water Data
Online portal at the following address: http://www.bom.gov.au/waterdata/. Catchment characteristics
(e.g. area) can be obtained from the Geofabric dataset available at the following address:
http://www.bom.gov.au/water/geofabric/. The source code used in this paper is available - instructions
for retrieving it are available from https://csiro-hydroinformatics.github.io/monthly-lstm-runoff/. The
code is made available under a CSIRO open-source software license for research purposes.




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





**Appendix**
This appendix includes reproductions of some of the report figures in which large outliers detract from
a decent visualisation of the bulk of the data points. Here the entire data set is included, whereas the
corresponding figures in the report are shown without the large outliers.

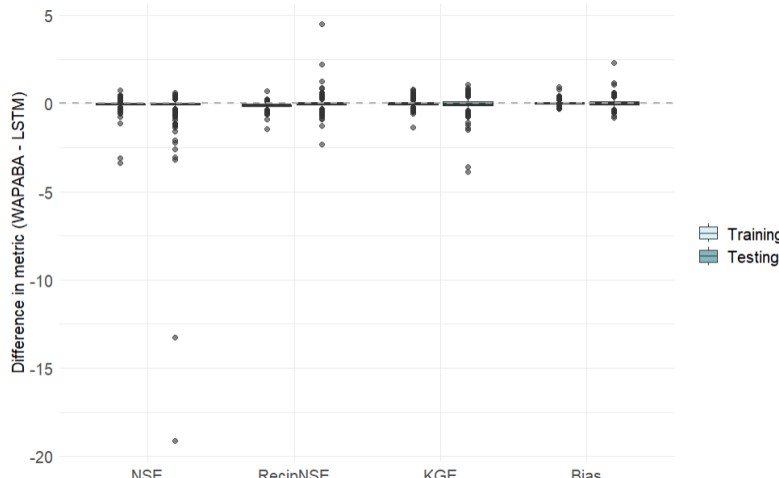


Figure A1: Difference in the metrics (WAPABA – LSTM) for each catchment. A reproduction of Figure 14 that includes
outliers.

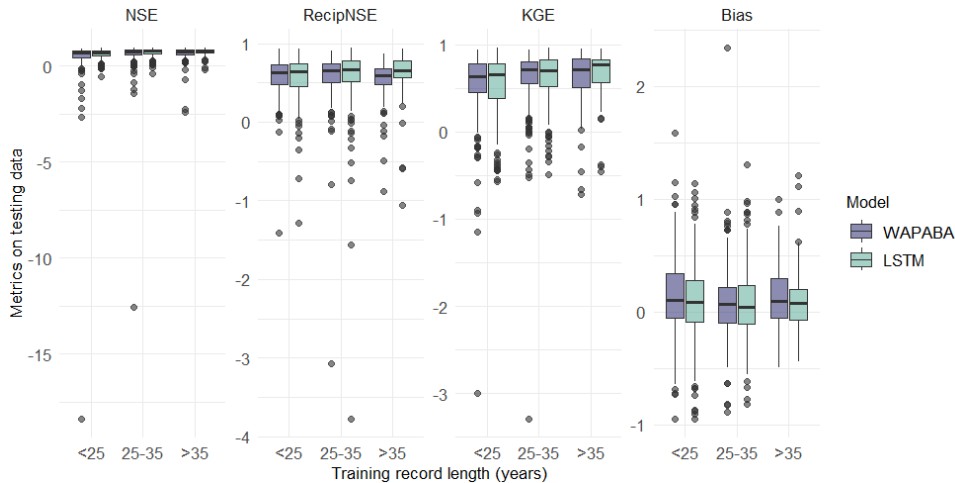


Figure A2: Effect of record length and training data size on prediction performance for each model type. A reproduction of
Figure 13 that includes outliers.