# Peer review of "Deep learning for monthly rainfall-runoff modelling: a large-sample comparison with conceptual models across Australia"

_Hydrology and Earth System Sciences, 2023_

## Author Response (AR1)

Editor comments:

I thank you for your timely replies in the discussion: both Referees have recommended major revisions and actually I think that you have not fully addressed all the main concerns raised by the two referees (due to the frantic holidays and post-holidays period, unfortunately they had no time to re-reply to your clarifications into the discussion, and I will certainly need their opinion if you decide to submit a revised version).

In particular, I fully support the request (made by both Referees) of a comparison with simpler, more widely-used machine-learning methods, and I believe that a comparison between individual feedforward networks and LSTMs is definitely inside the scope of this project and actually needed here. In fact, as you state earlier in your reply, your objective is "to help users of traditional modelling methods who are not necessarily experienced with machine learning understand what they might expect from running a very basic LSTM; the goal is not to maximise performance to cutting-edge machine learning standards." Indeed, non-expert users should not choose a more complex ML model (LSTM vs FFNN) if it is not necessary: FFNN have been used in hydrology (for modelling time-series) for more than 20 years, many hydrologists know very well how to use it and have described their best use in the literature: a comparison with the simpler and more known ML methods is necessary here, exactly to set an example for non-expert users, who should always start with the simpler structure (even if LSTM are more "fashionable" in these years, there is no reason to start directly from there, unless you demonstrate the benefit of their use, also in your comparison).

If you decide to submit a revised version, I would invite you to add this analysis, in addition to addressing all the other very relevant and constructive observations made by the Referees, in order to add depth and novelty to your work and make it even more significant for a vast audience of both conceptual and data-driven modellers.

Response: We thank you very much for your constructive comments and suggestions for improving our manuscript. As suggested, we have conducted a comparison with the more widely-known FFNN, and have added a discussion around this into the manuscript as well as comparison plots between FFNN/LSTM/WAPABA over the 496 catchments. This will hopefully allow researchers who have some familiarity with FFNNs to relate to this study. As per the remainder of the reviewers' comments, we believe the manuscript is now more comprehensive after including their suggestions on hybrid models, hyperparameters and other items. A careful proofreading of the entire document has also been performed, resulting in many minor corrections.

Reviewer #1 comments:

**Summary**
The paper compares a per-basin LSTM with a traditional RR model (WAPABA) on monthly data for basins in Australia and reports the LSTM to perform slightly better than WAPABA. The authors extensively investigate the results with respect to catchment size, flow conditions, and amount of data available.
I especially appreciated the detailed evaluation that breaks down the performance differences according to different possible causes.

My main concerns with this paper are related to methodology with the DL model. I believe that the presented comparison would have a different (likely more clear) outcome if the LSTM was trained differently. Please see my detailed comments below.

Response: We thank you for your time in reading our manuscript and providing comments especially with regards to the deep learning model methodology. Specific comments are addressed below. However, here we would like to re-emphasise that the objective of this study is to obtain the minimum performance level that a standard (non-expert) user might expect to get from off-the-shelf LSTM usage - to help users of traditional modelling methods who are not necessarily experienced with machine learning understand what they might expect from running a very basic LSTM; the goal is not to maximise performance to

cutting-edge machine learning standards. To emphasize this message more clearly, the following sentence has been added to the introduction to the paper:

> *'The goal here is not to maximise LSTM performance to cutting-edge machine learning standards, rather to ascertain the minimum performance level that a non-expert user might expect to obtain from basic usage of an LSTM with the input data regularly used in a conceptual model.'*

**Major Comments**

1. We know that single-basin LSTM RR models perform worse than globally trained ones. This has been very clearly shown for daily modeling [1] and I see no reason to believe this would be different for monthly data. In fact, I am quite confident that a well-trained global LSTM would outperform WAPABA more clearly than in the presented study. For instance, the authors state that their LSTM tends to underestimate high-flows (L639), which is exactly what global LSTMs are better at (because a high flow value in one basin is often not a very high value for another basin). There would also be fewer issues with the amount of training data, as the global model would have access to the samples from all basins at once. Another advantage of the global model is that one single model needs less compute to hyperparameter-tune and fit than 500 single-basin ones. The authors even discuss global models (L693) and their expected benefits (L689), so I don't understand why they wouldn't use one. If you think this requires a lot of coding work, I can recommend the NeuralHydrology library, which should allow to run your experiments with no or hardly any code modification (disclaimer: I'm one of the maintainers. This is just a suggestion, it's totally fine by me if you'd like to keep using your code).

Response: We are aware that global LSTM models have many benefits over individual models (for all the reasons stated above) and that a global model incorporating these ~500 stations would undoubtedly produce better LSTM results than we have obtained. One co-author is currently running a global model with NeuralHydrology on another project and we agree with the advantages outlined above. However, a global model was not chosen for this study as we endeavoured to make the comparison with individual-catchment WAPABA models as like-to-like as possible. This also provides the appropriate scale context for readers who seek to model a single catchment in their study, as is very often the case with traditional modelling studies. A frequently-heard reason why researchers do not attempt to use machine learning approaches is the small data size associated with individual catchment time series, and we were interested in demonstrating the lower limits of data availability required to fit an LSTM with individual catchment monthly data sets. Even though fitting a global LSTM over hundreds of catchments may lead to better results, in this study we have shown that similar performance to traditional models can be reached despite the fact that the LSTM was fit using limited data on a single catchment. A global LSTM is proposed for further work in a follow-on project.

2. Beyond the issue of how to train the ML model, I think it is questionable whether an LSTM is even the best choice of an ML model here. LSTMs are good for long input sequences with dependencies across many input steps, which is not the case here -- the paper ends up using just 6 time steps. These could easily be fed into a simple feedforward net (or even a random forest or an XGBoost model). Ideally, a paper that claims to investigate DL for monthly RR prediction should also check whether the LSTM is the right tool for this task. To be clear, it might be -- but it might also be no better or worse than a more lightweight and faster feedforward net.

Response:  To clarify the question of whether the LSTM is required in our case, or whether a simpler model would suffice, FFNNs have been run on the 496 study catchments and a comparison with LSTM and WAPABA results has been performed. The following text has been added, as well as comparison plots:

*"Though this study has focused on comparing the LSTM model to the WAPABA, readers may wonder if the more traditional feed-forward neural network (FFNN) may suffice in producing as good results. The FFNN has been used in hydrology for many years to model the relationship between climatic predictors and hydrological responses and many researchers are familiar with this basic neural network structure. However, the FFNN is a static network and does not consider the sequential nature of the input data. Though the six months of lagged predictor variables could be input as separate variables, this requires an increase in the complexity of the training space and is not likely to be the optimal choice for time series data as the cumulative impact of the predictor sequences may not be captured. Many studies have already considered the comparison of FFNNs to LSTMs for rainfall-runoff modelling and have determined the LSTM to provide superior runoff predictions (eg. Rahimzad et al., 2021). As an experiment, the FFNN has been run on this set of 496 catchments and added to the comparison of overall model performance, shown in Figure A3 of the Appendix. It can be seen that the FFNN leads to lower NSE, KGE, Reciprocal NSE, bias of the mean, bias of variability and correlation values, and therefore provides less accurate estimations of runoff than both the WAPABA and the LSTM. For this reason, the FFNN has not been included in the bulk of this study."*

3. The authors chose to use no validation period and justify this with an unreported "sensitivity test" (on one basin?). I do not find this convincing: it is unclear to me whether the test set remained untouched until final evaluation after hyperparameter tuning. However, in this case HP-tuning only happened on a single basin (which is far from ideal in itself), which means that at least most of the test set was apparently not touched for validation. Still, I would prefer to see a separate validation period. If lack of data is a concern, the authors could opt for a cross-validation scheme for HP-tuning.

Response: The omission of a validation period was a deliberate choice based on the like for like comparison we are doing. We wanted to train on the same data as WAPABA and predict similarly, with identical duration and dataset size. As mentioned earlier, the objective was not to determine the optimal model configuration, rather to use a vanilla setup. This approach was followed with the selection of hyperparameters, with a set chosen that yielded reasonable results on a range of catchments. The testing set remained completely untouched until final evaluation. Data-leakage between the training and testing sets was avoided by splitting the training set for the sensitivity test so that 80% of the training set was used for training and 20% of the training set was used for validation to monitor for over-fitting. The testing set was not used at all during the sensitivity test. Hyperparameters were tuned in a separate process on a subset of catchments, also using only the training set.

4. Open research and reproducibility:
   1. I was unable to find the actual code and configuration files under the link that is supposed to provide the source code for the paper's experiments. All I found is a notebook with a toy example.
   2. I would appreciate the authors to provide a ready-made download link to the forcings and streamflow data, rather than pointers to several government sites that leave people to figure out how to find the data from ~500 basins themselves. A single zenodo link would be far easier. If that's not possible (e.g., for license reasons), please provide a script to download the data (and to put it in the correct format if any changes are needed).

Response: It is our aim to ensure the data and code are accessible. This section now reads:

*'All data used in this paper are accessible through the website of the Australian Bureau of Meteorology. Rainfall and potential evapotranspiration can be downloaded from the Australian Water Outlook portal at the following address: https://awo.bom.gov.au/.*

*Streamflow can be downloaded from the Water Data Online portal at the following address: http://www.bom.gov.au/waterdata/. Catchment characteristics (e.g. area) can be obtained from the Geofabric dataset available at the following address: http://www.bom.gov.au/water/geofabric/. The deep learning source code used in this paper is available at: https://csiro-hydroinformatics.github.io/monthly-lstm-runoff/ including an overview and instructions for retrieving the source code and setting up batch calibrations on a Linux cluster. The code is made available under a CSIRO open-source software license for research purposes.'*

**Minor Comments**

1. Overall, and especially in section 3, I think there is too much focus on the performance difference between train and test period. This is not really meaningful for ML models which can easily be trained to NSE >> 0.9 on the training set and still perform well on test data.

   - This is a good point. However, what is interesting in these plots is the comparison between WAPABA and LSTM as well as the training and testing periods. For example, in Figure 4 the difference between WAPABA and LSTM performance is relatively large during the training period but similar during testing, indicating perhaps a higher tendency towards overfitting by the ML models than traditional modellers would be expecting. In the boxplots in Figures 4, 5 and 6, the greater depth of the testing boxes indicates a greater spread of values than seen during training, which is discussed in the text.

2. I would transpose table 3 and add columns for the different KGE components.

   - Table 3 has been transposed and the KGE components added as suggested.

3. Fig 10: I might be misunderstanding something, but what is your definition for "no flow"? Apparently it is not Q = 0, because the observation axis still shows variation.

   - Thank you for pointing this out as it will likely be confusing for others as well. As the data has been standardized by catchment, Q=0 does not end up exactly at zero for each catchment. An updated explanation has been added in the text and figure caption:

   *'For comparison purposes in this section, the raw observed and modelled flow data are standardized by station based on the mean and standard deviation of all observations at that station during the study period. The observed mean is subtracted from each value before dividing by the standard deviation of the observations, allowing for basins with a range of flow volumes to be compared.'*

   *and*

   *'Note that the standardization procedure used in this section leads to standardized 'no-flow' data points that do not fall exactly on zero in the plot even though the raw flow values at these points are zero.'*

   *In the caption: '… the data have been standardized based on observed mean and standard deviation leading to non-zero values in the 'no-flow' category.'*

4. Fig. 12: Would be interesting to look at the observed and predicted hydrographs of the one basin where LSTM is poor and WAPABA is good.

- This might indeed be of interest to explore further. However, as the manuscript is already on the long side with the existing investigations into many subgroups, we have decided that including an extra figure pertaining to specific conditions at one catchment would not add enough value to the overall study. This catchment has barely squeaked into the LSTM NSE<0 category, as shown below, and the text has been updated to include a bit more information for the curious reader:

*"in this catchment the LSTM prediction is on the border of poor and fair (NSE=0.001)"*

[Figure]

5. L42 "In some cases...": I think this formulation is a bit too weak.

    - This wording has been removed.

6. L52: The citations should include the first LSTM paper [3] and L54 should include the paper that showed how to train global LSTMs at CONUS scale [4].

    - These citations have been added as suggested.

7. L58: I'd add [2] here, which I think is a bit closer to the content of the sentence than the other two citations.

    - Citation has been added as suggested.

8. L83: There are many more, I suggest adding an "e.g.," to the list of citations

    - 'eg.' has been added as suggested.

**Typos**

1. L279 model's

    - Corrected.

2. L619 broken reference

    - Corrected.

3. Several incorrect uses of \citep vs. \citet

    - Issues have been corrected where found.

Reviewer #2:

The paper compares the long short-term memory model (LSTM), a type of recurrent neural network, to a well-known conceptual rainfall-runoff model (WAPABA) using monthly data for basins in Australia. They noted that LSTM is just as effective as conceptual models for simulating daily runoff in various regions of the world. Due to the significance of monthly data in water resources planning, they studied the LSTM while establishing a monthly rainfall-runoff relationship, unlike earlier studies. In their comprehensive work, the ability of LSTM to produce an accurate monthly runoff simulation was tested under different conditions across the Australian continent. During this pathway, they analyzed various indices and concluded that the LSTM outperforms WAPABA in the majority of catchments. Even though a great deal of effort went into the calculations, my concerns have nothing to do with the utilization of LSTM (tuning the hyper-parameters

regarding its internal architecture, training algorithm, etc.). Consequently, I suppose that considering my remarks below can raise the scope of the study.

Response: Thank you for your comments on our manuscript especially relating to the integration of machine learning and traditional models in hybrid configurations. These considerations will strengthen the message of our work and tie it into broader contemporary themes in hydrological machine learning.

Major Comments:

1. As they specified, the accuracy of data-driven models trained for runoff simulation is heavily dependent on the quantity of lagging data, and as a result of numerous trials, they decided on a lag of six months. In this case, there are two issues that need to be discussed in the paper. First, are we confident that the LSTM can substitute for a conceptual model since it is so dependent on antecedent data? Additionally, as discussed by Robertson et al. (2013), when a catchment is wet, antecedent runoff does not promptly respond to antecedent precipitation; rather, soil moisture and groundwater storages mostly refill. Under these circumstances, antecedent runoff values may underestimate the actual soil moisture conditions, resulting in relatively low runoff simulations. Other than the median flow (i.e., low and high flow), might this explain the differences between the simulations at different percentiles?

Response: This is an interesting question. The advantage of the LSTM, in terms of the choice of possible machine learning models to use, is that the LSTM processes antecedent data to make predictions at the current timestep. Therefore, to predict the runoff at time t, our model is considering the precipitation, evapotranspiration etc. sequentially over the previous 6 months (t-6). In this way the model will learn patterns such that periods of dryness followed by precipitation may lead to less runoff (due to refilling of soil moisture and groundwater storages) than precipitation occurring at the end of a long, wet period. A lag time of 6 months was determined by trial and error to produce the best predictions and this is supported by hydrologic knowledge that catchment conditions longer than 6 months in the past would likely have little effect on current runoff.

2. In addition, it may be conceivable for another machine learning model (standard feed forward neural networks, support vector regression etc.) to supersede LSTM, particularly for the runoff simulation using 6-month lagged data. In this sense, I also strongly recommend conventional machine learning models against WAPABA. Otherwise, we will just be exploring a fiction revolving around the increasingly popular use of deep learning to monthly runoff data.

Response:   It is possible that a feedforward neural network (FFNN) may be used to model the system, however this structure does not embody the sequential nature of the data, requires an increase in the complexity of the training space and is not likely to be the optimal choice for time series data. To clarify the question of whether the LSTM is required in our case, FFNNs have been run on the 496 study catchments and a comparison with LSTM and WAPABA results has been performed. Please see the response to Reviewer 1 for further details.

3. Another issue to be discussed in the paper is the coupled conceptual-machine learning modeling framework. Although several references are provided in Section 4, it is important to clarify what benefits this hybridization brings and what limitations of individual machine learning models it can address. In fact, soil moisture and groundwater recharge outputs derived from calibrated WAPABA model are likely to strengthen the predictors of LSTM.

   Moreover, in a rather limited number of studies, the internal structure of conceptual hydrological models has been replaced with machine learning approaches (see Okkan et al., 2021). Employing LSTM to modify the internal runoff partitioning mechanism of WAPABA is not deemed required in this study. But it would be appropriate to refer to and discuss this kind of nested hybridization in Section 4, just to shed light on potential future studies.

Response: Thank you for highlighting this as an important point. We have touched upon this topic in the Discussion section, however it would be interesting to expand the discussion to include the benefits of hybridization in overcoming limitations of these individual machine learning models. The following text has been added to the 'future work' section of the Discussion to point the reader to some recent hybrid models in the literature and describe what could be done in the case of this study:

> *'Another consideration may be hybrid modelling frameworks, which combine aspects of conceptual models with machine learning models. These have the potential to draw benefits from both types of models to produce more interpretable and possibly more physically realistic predictions. By leveraging the particular skills of each model type, the limitations inherent in each may be reduced. For example, Okkan et al., (2021) embedded machine learning models into the internal structure of a conceptual model, calibrating both the host and source models simultaneously, and found the product outperformed each model individually. Li et al., (2023) replaced a set of internal modules of a physical model with embedded neural networks, leading to improved interpretability as well as predictions that are comparable to pure deep learning (LSTM) predictions. The authors found that replacing any of the internal modules improved performance of the process-based model. In the Australian context, Kapoor et al., (2023) studied the use of deep learning components in the form of LSTMs and convolutional neural networks to represent subprocesses in the GR4J rainfall-runoff conceptual model for a set of over 200 basins. It was found the hybrid models outperformed the conceptual model as well as the deep learning models when used separately, and provided improved interpretability, better generalisation and an improvement in prediction performance in arid catchments. In this case of this study, the soil moisture and groundwater recharge outputs derived from the WAPABA model would likely be useful as additional predictors for the LSTM model. '*

Minor Comments:

1. The figures for grading criteria are appealing, but it would be useful to visualize them on the continent using a GIS tool.

   - Thank you for this suggestion. Before the original submission we had tried a GIS visualisation of the results and found that the geographic visualisation was not effective enough in conveying the overall message to warrant inclusion in the manuscript. The plots that have been included were chosen to specifically convey the results of each comparison.

2. The way of citation in some lines does not seem usual (e.g., L83).

   - In this case, the geographic location of each study is important to include with each reference. This is the only location in the manuscript where references are reported like this.

3. It should be checked whether the unit of "Inverse K" in Table 2 is mm/day or mm/month.

   - This has been checked and the unit of 1/day as currently in the table is correct.

4. There is no identity in the presentation of references. Some journal names are abbreviated, some are not.

   - Journal names have been checked and abbreviated names have been expanded.

5. Line 619 has the following statement: "Error! Reference source not found"

- Corrected.

6. It would be appropriate to give the conceptual diagram of WAPABA.

- A diagram of the WAPABA model has been added to the methods section.

7. Also, was the warm-up period used while applying WAPABA? To avoid any bias associated with initial storage values, this can be useful.

- The model was warmed up 2 years prior for calibration. This information has been added to the methods section: *"The model was calibrated with a warm-up period of 2 years to avoid possible bias associated with initial values."*

---

## Referee Report (RR1)

I appreciate the authors' explanations and their effort to add a comparison to FF networks. Apart from that, not too much has changed in the manuscript, so I don't have a lot of comments. You will not be surprised that I continue to be unhappy about the LSTM setup. To me, the framing of investigating how well you do if you don't follow the best practice is not entirely convincing. Shouldn't we rather educate the "non-expert user" about the correct modeling setup? Wouldn't that user be interested in how much better their model could be, especially in an investigation of "the lower limits of data availability"?

The small data size argument is indeed frequently heard, but it is largely a fallacy. E.g., for Australia, there exists plenty of data (Camels-AUS, Caravan, or the authors' data set) which anyone can use in conjunction with their individual catchment.

Overall, I feel that in its current form, the paper becomes one of the many many papers that compare single-basin LSTM vs. conceptual model X on gauge set Y. Nothing about it is *wrong* (in fact, this is one of the much better ones because |Y| >> 1), but it also doesn't show the full picture, and I fear that adding to this type of papers will only further distort the picture of LSTM-based modeling in the literature.

---

## Author Response (AR2)

Dear Elena,

We are very pleased that this paper has been accepted for publication with HESS.

In this final resubmission, I hope it is ok to change one word of the title – from 'classical' to 'conceptual'. We believe this change to the more specific wording will make the paper more easily found by researchers who are searching for comparisons between machine learning and more traditional approaches.

I have also updated a few of the captions to reflect the (a), (b) annotations that have been added to the figures.

Kind regards,

Stephanie